# A Hash-Based RFID Authentication Mechanism for Context-Aware Management in IoT-Based Multimedia Systems

**DOI:** 10.3390/s19183821

**Published:** 2019-09-04

**Authors:** Deebak B D, Fadi Al-Turjman, Leonardo Mostarda

**Affiliations:** 1Schoool of Computer Science and Engineering, Vellore Institute of Technology, Vellore-632014, India; 2Computer Engineering Department, Antalya Bilim University, 07190-Antalya, Turkey; 3Computer Science Department, Camerino University, 62032-Camerino, Italy

**Keywords:** multimedia device management systems, RFID, context aware sensor management systems, replay, de-synchronization, traceability

## Abstract

With the technological advances in the areas of Machine-To-Machine (M2M) and Device-To-Device (D2D) communication, various smart computing devices now integrate a set of multimedia sensors such as accelerometers, barometers, cameras, fingerprint sensors, gestures, iris scanners, etc., to infer the environmental status. These devices are generally identified using radio-frequency identification (RFID) to transfer the collected data to other local or remote objects over a geographical location. To enable automatic data collection and transition, a valid RFID embedded object is highly recommended. It is used to authorize the devices at various communication phases. In smart application devices, RFID-based authentication is enabled to provide short-range operation. On the other hand, it does not require the communication device to be in line-of-sight to gain server access like bar-code systems. However, in existing authentication schemes, an adversary may capture private user data to create a forgery problem. Also, another issue is the high computation cost. Thus, several studies have addressed the usage of context-aware authentication schemes for multimedia device management systems. The security objective is to determine the user authenticity in order to withhold the eavesdropping and tracing. Lately, RFID has played a significant for the context-aware sensor management systems (CASMS) as it can reduce the complexity of the sensor systems, it can be available in access control, sensor monitoring, real time inventory and security-aware management systems. Lately, this technology has opened up its wings for CASMS, where the challenging issues are tag-anonymity, mutual authentication and untraceability. Thus, this paper proposes a secure hash-based RFID mechanism for CASMS. This proposed protocol is based on the hash operation with the synchronized secret session-key to withstand any attacks, such as desynchronization, replay and man-in-the-middle. Importantly, the security and performance analysis proves that the proposed hash-based protocol achieves better security and performance efficiencies than other related schemes. From the simulation results, it is observed that the proposed scheme is secure, robust and less expensive while achieving better communication metrics such as packet delivery ratio, end-to-end delay and throughput rate.

## 1. Introduction

In the recent past, the development of smart computing devices in the Internet of Things (IoT) has grown exponentially [1,2]. These devices share the geo-location of physical objects with other systems. They are widely used in smart infrastructure to build the smart city features such as smart eHealthcare, finance, e-Governance, parking and transportation [3]. There are more physical objects involved in connecting IoT-based applications in smart infrastructure. This infrastructure often integrates radio frequency identification (RFID) to configure different application environments such as smart inventory, toll-booth collection, object identification, anti-counterfeit protection and the automobile industries [4]. In contrast to bar-coding systems, the physical objects equipped with an RFID tag are not required to be in line-of-sight to read the encapsulated data. As a result, a long-sighted RFID-tag position can easily extract a large amount of RFID information to replace traditional supply-chain management systems with an RFID-based authentication system. Besides, the real-time objects track and identify the RFID information wirelessly to automate communication systems. It is considered as a promising technology to monitor the smart objects.

The technology known as Radio Frequency Identification (RFID) reads the physical objects and automatically recognizes the relative object details, i.e., it is basically a non-contract recognition technique [5]. This technique uses some type of artificial inference to sense the radio frequency that provides a communication between the tags attaching with the objects and the readers connecting with the backend server systems. Using this technology, several application systems such as chain management, credit-card, electronic passport verification, vehicle systems (i.e., charging and keyless entry), etc. have been designed and developed. Specifically, the countries like Japan, USA and other developing countries are nowadays becoming equipped with advanced RFID systems [6]. Lately, it has undergone further advancement in the form of electric induction [7] that recognizes the tag attachment object to read the object information. 

RFID tags do not need any light source to sense the data through external materials, which makes them durable, low-cost, reliable and secure in comparison with bar-code systems [8]. Most retailers have employed this technology for integration of tag attachments that allow an authorized dealership to deliver the goods. For instance, Wal-Mart imbeds RFID tags in products which reduces the manpower and the materials resource needed by producers [9]. Moreover, this technology is now considered to be an integral part of people’s daily activities such as the use of cellphones, automobiles, household objects, etc. It does not need any physical contact to sense or scan different types of objects. Importantly, it uses one signal to scan the various types of barcodes and in addition it has an ability to read and write tags multiple times [10]. This technology can even be used in different climate conditions like snow, fog and in packaging [11].

It is perceived to be a significant advancement for the development of future markets. Most of the enterprises and manufacturing industries including governments, banking, transportation, agriculture, food safety, healthcare, etc. are attaching these tags to automate the product delivery process faster in order to improve customer service and business automation efficiency. In the past, the usage of RFID, specifically in the range of high-frequency (HF-13.56 MHz) has gained much attention. Particularly, the Near Field Communication (NFC) standard has been designed and improved for the five types of NFC’s i.e., in correspondence with different types of ISO/IEC and JIS standards. In general, the type 1, 2 and 4 are overlaid for ISO/IEC 14443-A, whereas the type 3 and 5 are dealt with in JIS X6319 and ISO/IEC 15693 (18000-3), respectively [12]. New manufacturers like NXP Semiconductor (Newburyport, MA, USA) have advanced the use of NFC technology, i.e., ntag-213/215/216. NFC cards are used in access control to improve the enterprise security. 

Smartphone usage has developed exponentially in the past few years. As referred in [13], the number of smartphones used worldwide is predicted to be 4.49 billion (i.e., 59.9% of the global population). This generalization demands high user-level confidentiality to ensure security and privacy. In the USA [14] and EU [15], national program and strategies have intensified researchers’ attention in support of privacy. Recently, the computation power of programmable smartcards has increased tremendously for the massive development of smart electronic devices such as public transportation, e-passports, e-ticketing and e-identification. These RFID-based communication devices are expected to provide tag anonymity in order to ensure privacy enhancement. Due to security and privacy issues, RFID-based authentication schemes are gradually becoming assimilated in several real-time applications. 

Generally speaking, RFID manages to track the unauthorized client access to ensure the privacy of RFID tags to resist potential vulnerabilities. Due to their limited computation resources, power and storage capabilities, it is more complicated to apply the expensive cryptographic operations in low-power RFID systems. Moreover, the expensive computations have slowed the development of RFID technologies [16,17,18]. Most significantly, adoptive RFID technologies cannot properly exploit the authentication process to enhance their security-level. In order to restrict malicious activities, the source messages should be prevented from fake broadcasting. It is noted that the physical security of RFID tags should be proactively secured to prevent unauthorized access. Moreover, the attackers then would not be able to track the previous user tag information to impersonate a legitimate user. Thus, this paper introduces a novel hash-based RFID mechanism using synchronized secret session key value and its objective is to: (1) Provide user privacy; (2) Reduce the power consumption; (3) Reduces cost of the tag; and (4) Offer mutual authentication, tag-anonymity and traceability.

The remaining sections of this paper are organized as follows: Section 2 discusses the research background, including communication models, motivation and enabling technologies and its key objectives; Section 3 presents context-aware sensor management systems; Section 4 proposes a novel hash-based RFID mutual authentication protocol; Section 5 presents an informal security and performance analysis; Section 6 demonstrates the experimental analysis using NS-3; and finally Section 7 concludes the research work.

## 2. System Models

This section discusses the system communication models and research motivation to explain the key factors of hash-based RFID authentication. 

### 2.1. System Communication Model

A basic RFID system consists of three communication entities, namely the receiver (reader), transponder (tag) and a backend database such as a server to store and analyze the data. In general, the RFID tags are attached on physical objects that locate and identify physical things among thousands of objects. Each RFID has a small built-in antenna that attaches a microchip with limited memory space to store the identities of data objects [19]. The RFID reader is basically used as a scanner that interrogates the tag information existing in scanning system environment. A server such as a backend database strategically handles the massive amount of generated data for data processing and storage, offering an influential and necessary processing capability and storage space. Furthermore, it operates the system processor to read, manage, control and store the tag data from the attached tags in the RFID reader as illustrated in Figure 1.

### 2.2. Research Motivation

RFID is used to collect the static informational data; however tags cannot extract dynamic informational data, like humidity, acceleration and temperature. On the other hand, sensor systems nowadays are capable to collect all kind of dynamic information data to fill communication gaps using RFID systems. Therefore, the RFID system is integrated with the sensors to collect and monitor the environmental status related to the informational data items. This research refers to a device that integrates a RFID tag and a sensor called Sensing RFID (SRFID). The system device tries to integrate the sensor into an RFID tag to deal with the several challenging issues in large-scale or open-loop systems. The first challenge of the system is the use of non-smart sensors that have specific application requirements for different types of sensors. Moreover, the system models and its related sensor parameters of devices produced by different manufacturers have different types of functionality, even if the sensors are of the same nature. 

The following issues are identified, as and when the sensors of the SRFID are configured in a standard way for off-the-shelf readers: (1) SRFID has unique differentiation by the readers; (2) The sensor parameters have automatic sensor recognition features; and (3) The parameters related to the sensor have effective configurations. When the sensor collects the sampling datasets to transfer them to the reader, the datasets, such as parsing and identification, represent additional problems. The identification problems are related to the description of the sensors. The objective of the sensors is to provide interoperability between SRFID tags and readers/host applications, though general purpose sensors, such as analog and digital devices do not have such self-descriptive functions and capability to define the sensors. The second challenge of the system is the data storage related to the SRFID tag. The sensors are diverse in nature, and thus may vary for every SRFID tag. Therefore, the data storage method needs features such as simplicity, flexibility and efficiency. The reader interfaces the various sensors with the application programming interface (API) to access and control all related data of the SRFID tag. Besides, this data storage method allows convenient access to append/remove the sensor from the tag. 

The third challenge of the system is the sensor sampling mechanism and it is used to set up an operational mechanism to possess the needed functions to configure, collect and parse the sampling data. To overcome the above challenging problems, this paper focuses on the integral components of an active SRFID tag, namely a self-descriptive sensor, a data storage sensor and an operational sampling sensor. In addition, this paper is restricted to the self-descriptive sensor in consideration of the Plug and Play (PnP) feature [20,21]. In RFID, the issues such as security and privacy are primarily focused on physical [22] as well as security authentication mechanisms [23,24,25]. As the physical mechanism incurs more additional cost, security authentication schemes are very attractive in practical use. Chien et al. [23], Kim et al. [24,25] and Hajny et al. [12] have studied cross-examination. Since the authentication schemes [12,23,24,25] do not validate the secret session key of the tag and reader, they are susceptible to various kinds of malicious attacks, such as eavesdropping and tracing, replay, man-in-the-middle and desynchronization. Besides, they do not comply with the security properties of tag anonymity and untraceability.

### 2.3. Enabling Technologies and Its Key Objectives

The Internet has become more prevalent for social-media networks and various emerging technologies such as wireless sensor networks (WSN), Internet of Things (IoT), cloud computing (CC) and big-data management. These technologies enable people to communicate and share their interests in several ways. As a result, technological advances create new application models and business opportunities to offer comfort, safety, and more computational efficiency. Of late, CC and IoT have been more relevant for industry and academic collaboration. Ashton presented the IoT concept [26] that defines the purpose of physical objects and wireless channels [27]. McCarthy [28] introduced the cloud computing concept that derives a large-scale distributed system to drive several economic benefits such as virtualization, data storage, and computation power. 

IoT and CC are service computing platforms that allow object interconnection, including personal and sensitive data, over wireless channels [29,30,31]. The collective data of physical objects or sensors is generally stored on a cloud-server [32,33,34,35]. However, features such as security and privacy are highly demanded to prevent malicious activities [36]. Ferrag et al. [37] and El-Hajj et al. [38] discussed the IoT requirements, including authentication, authorization, confidentiality, privacy and message integrity to signify the importance of node protection. Therefore, to fulfill the standard requirements of an authentication protocol, a suitable mechanism is highly recommended. It can verify the user identities to determine whether he/she is vulnerable or not on open networks [37,38]. Due to network vulnerabilities and Internet security, user authentication plays a crucial role [39]. 

From the above discussion, the important aspects of RFID authentication protocols have been studied for different IoT environments including industrial management, payment schedules, and several emergency systems. These environments use RFID-tags to achieve more computation power, speed, and physical object robustness than traditional barcode systems. Moreover, technological advancements have addressed several security issues for RFID-based security systems. Concerning the property of untraceability, several RFID-based authentication protocols have simply traded the privacy for the purpose of better system performance. Most of the authentication protocols merely encrypt the tag identity of RFID using cryptographic functions. In case of verification, the reader or back-end server is expected to perform an extensive operation to authorize the RFID-tag, resulting in poor system performance. 

Of late, several authentication protocols have been proposed for low-cost RFID systems [40,41,42,43,44,45,46,47]. However, they are reported to have high execution cost, security weaknesses, and vulnerabilities. Chien et al. [40] presented a strong authentication and strong integrity (SASI) protocol that is based on ultra-lightweight authentication. However, their protocol is highly prone to tag tracing and desynchronization attacks [41,42,43]. To address the critical issues of SASI, Peris-Lopez et al. [44] introduced the Gossamer protocol. Unfortunately, it could not resist desynchronization attacks [45]. Of late, Fan et al. [46] presented an ultra-lightweight authentication for mobile commerce applications including cryptographic operations such as XOR, shift and addition modulo operations, but Aghili and Mala [47] have proven that the Fan et al. scheme cannot resist physical and reader impersonation attacks. Table 1 summarizes the challenging issues of existing RFID-based authentication protocols. 

To address the above issues, this paper presents a novel hash-based RFID mechanism using synchronized secret session key values. This mechanism tactfully meets the crucial security requirements of IoT-based multimedia systems, namely mutual authentication, untraceability, and resilience to desynchronization attacks. Moreover, it has a built-in context aware management system to handle the storage parameters and thus it can reduce the additional communication cost to improve the overall system performance. Importantly, it does not perform any exhaustive search to affect the execution of back-end servers.

## 3. Context-Aware Sensor Management Systems

Figure 2 shows the block diagram of the integration of a virtual TEDS system in IEEE 1451 [53]. The purpose of IEEE 1451 is to identify the parameters of non-smart sensors to access the memory where the datasets related to the TEDS are stored. The IEEE standards, namely IEEE 1451.7 [54], ISO/IEC 24753 [55] and ISO/IEC/IEEE 21451-7 [55] integrate RFID tags and sensors to specify and interface the sensor security and data structure. The standard of ISO/IEC/IEEE 24753.7 is identical to the IEEE standard of 1451.7 which specifies the model related to the application interface to integrate the RFID tag and sensor. This integration is done to execute the functional commands of the sensor application system. These standards integrate the RFID tag and sensor to develop a smart sensor system with RFID tags. However, they do not work with the integral components of existing sensors, such as analog and digital ones to act as a smart sensor system with RFID tags. 

The sensor management system has a self-descriptive model like Sensor Web Enablement (SWE) [56,57,58,59] initiated by the Open Geospatial Consortium (OGC) to standardize the data encoding and web service interface. In addition, the OGC has built a web-based sensor model for the Sensor Modeling language (SensorML) [21]. The W3C Semantic Sensor Network (SSN) group has proposed an ontology description of SSN [22]. However, these data descriptive sensors are generally employed at the host computer/network level but not the sensor-node level. Moreover, these data descriptive methods use Extensible Markup Language (XML) that has an enormous functional format and thus this language requires an XML-Parser to consume less memory storage. 

Therefore, XML files are difficult to deploy in self-descriptive/context-ware sensors to retrieve, remove and modify the sensing data. The self-descriptive/context-aware method is usually not suitable for sensor application systems, and so this paper proposes a novel strategy to back up the sensing data in the SRFID tags to protect the exchanged information. 

## 4. Novel Hash-Based RFID Mutual Authentication Protocol

A novel hash-based RFID mutual authentication protocol using a secret-session key is proposed. A novel session-key sharing strategy is used to secure the communication between the reader/user and the back-end database server. 

Figure 3 illustrates the novel hash-based RFID mechanism. The phases of the mechanism, namely Pre-phase registration, Readers Pro-Tag request and Response, Tag Mutual Session-key Authentication, Back-end Server Key Authentication and Session-key Updating are executed to solve the challenging issues of the existing protocols, such as security, privacy and forgery. To address the issue of security, this paper uses a secret-session key as a significant feature. In addition, the session keys are subsequently generated at the back-end database server ${\mathrm{SS}}_{k}\to {\mathrm{SS}}_{k-1}$ to update the output values of the session keys of the tag. Table 2 shows the important notations used in this paper.

### 4.1. Phase I: Pre-Phase Registration

The execution flow of the pre-phase registration are as follows:
(1)The back-end database server and the tag mutually share their credentials, such as Tag-$I_{D}:{\;I}_{D_{K}},\;\mathrm{one}–\mathrm{way}\;\mathrm{hashing},\;\mathrm{secret}-\mathrm{session}\;\mathrm{key}:{\;\mathrm{SS}}_{k}$ in the pre-phase registration of the proposed mechanism.(2)The reader and tag have a unique random number generator to authenticate the services like role assignment. For each tag, the back-end database server collects the parameters, namely $I_{D_{K}},{\;\mathrm{SS}}_{k},{\;\mathrm{SS}}_{k-1}$ and assign the values.(3)${\mathrm{SS}}_{k}$ is the secret-session key of the current session tag k.(4)${\mathrm{SS}}_{k-1}$ is the secret-session key of the previous session tag k−1, since its initial value is set to null.(5)Data: Information of the object/role that is tagged to the back-end database server to assign the access privileges. 


### 4.2. Phase II: Readers Pro-Tag Request and Response


(1)The reader randomly selects an integer $R_{i}$ to send a request to the tag.(2)Then, the tag generates a random integer $R_{j}$ to compute: $X=H\left({\mathrm{SS}}_{k}∥I_{D_{K}}\right)\;⨁{\;R}_{i}$; $Y=X\;⨁\;H\left(I_{D_{K}}∥R_{i}∥R_{j}\right)$; and $Z=X\;⨁\;H\left(Y\;⨁{\;T}_{s1}⨁R_{i}\right)$, where $T_{s1}$ is the current timestamp of the tag.(3)After the computation of {X, Y,Z}, the tag sends its response message as $\left\{X,\;Z,{\;T}_{s1}\right\}$ to the reader.(4)After receiving the response message $\left\{X,\;Z,{\;T}_{s1}\right\}$ from the reader, the reader sends the message of response $\left\{X,\;Z,{\;T}_{s1},R_{i}\right\}$ to the back-end server, after being added to the computational integer of $R_{i}$.


### 4.3. Phase III: Tag Mutual Session-Key Authentication

Upon receiving/extracting the tag informational data from the database, the back-end server executes the computational following steps:
(1)If $\left(T_{s1}-T_{s2}\right)>\Delta T_{s}$, then the back-end server terminates the login request of the user, where $T_{s2}$ is the current timestamp in the remote-server and $\Delta T_{s}$ is the expected transmission delay.(2)Compute: $R_{j}^{*}=X⨁H\left({\mathrm{SS}}_{k}∥I_{D_{K}}\right)$(3)Validate: $Z^{*}=H\left(A⨁H\left(I_{D_{K}}∥R_{i}∥R_{j}^{*}∥\right)⨁R_{j}^{*}⨁T_{s2}\right)\approx Z$(4)Repeat the execution steps (1) and (5) till the value of $Z_{i}^{*}$ is equal to $Z_{i}$ from the response message of reader. If the values are equaled, then the right tag will be found.(5)Compute: $U=H\left(X⨁H\left(I_{D_{K}}∥R_{i}∥R_{j}\right)⨁T_{s2}⨁{\mathrm{SS}}_{k}\right)$


Upon receiving the appropriate-tag, the back-end server communicates the tag information to the readers to update the value of secret session-key. In addition to the computation and validation, this phase also executes the following steps to update the readers’ session-key:
(1)The back-end server forms a new message format as ${\mathrm{New}}_{\mathrm{msg}}=\left\{\mathrm{Data}∥U\right\}$, where Data is the information of the tag to be communicated to the reader to ensure the property of mutual authenticity. If the back-end server fails to deduce a valid right tag, then the server deduces that there is an invalid message authentication to terminate the user session. After the execution of new message ${\mathrm{New}}_{\mathrm{msg}}$, the reader executes the following steps to maintain the communication:(2)The back-end server sends the new message of ${\mathrm{New}}_{\mathrm{msg}}=\left\{\mathrm{Data}∥U\right\}$ to the reader.(3)Upon receiving the new message of ${\mathrm{New}}_{\mathrm{msg}}$ from the back-end server, the reader excludes the parameter of Data and sends the parameter of U to the tag to maintain the communication.


### 4.4. Phase IV: Back-End Server Key Authentication and Session-Key Updation

Based on the message U at $T_{s3}$, the tag executes the following steps to authenticate the back-end server:
(1)If $\left(T_{s3}-T_{s2}\right)>\Delta T_{s}$, then the tag terminates the login request of the user, where $T_{s3}$ is the current timestamp in the tag and $\Delta T_{s}$ is the expected transmission delay.(2)Compute: $U^{*}=H\left(Y⨁{\mathrm{SS}}_{k}⨁T_{s3}\right)\approx U$, if the condition is valid, then the tag authenticates the back-end server to confirm that the computed hash value is identical to incur the value of U from the reader.


After the authentication of back-end server, the secret session-key ${\mathrm{SS}}_{k}$ is updated into ${\mathrm{SS}}_{k+1}$ to server the communication between server and tag.

## 5. Security and Efficiency Analysis

This section focuses its discussion on the security and performance analysis by comparing the security properties and computation cost with other related protocols [12,24,25].

### 5.1. BAN Logic Analysis

BAN logic [60] analysis is highly preferred to signify the design process that ensures the security structure of authentication protocol when it starts to build [61]. Moreover, it is a standard way to satisfy the security features of any application system [62,63]. In general, the formal analysis model is comprised of four methods: logical procedure, common analysis, model detection and proof of theorem. This paper chooses BAN logic to verify a belief of agreements as a logical structure [64] in order to analyze the security features of proposed hash-based protocol. The proof of verification is classified into four descriptive parts that are as follows:

A) Protocol Explanation

In this part, the information processes and its related transmission parameters are briefly introduced to entail the structure of the systems where $T_{R}$ represents the RFID tag, $R_{H}$ represents the reader and $B_{S}$ represents the back-end server:
(1)$R_{H}\to T_{R}:\;\left\{R_{i}\right\}$(2)$T_{R}\;\to {\;R}_{H}:\;\left\{X,\;Z,{\;T}_{s1}\right\}$(3)$R_{H}\to B_{S}:\;\left\{X,\;Z,{\;T}_{s1},R_{i}\right\}$(4)$B_{S}\to R_{H}:\;\left\{E\right\}$, where E=Data∥U(5)$R_{H}\to T_{R}:\;\left\{U\right\}$, where $U=H\left(X⨁H\left(I_{D_{k}}∥R_{i}∥R_{j}\right)⨁T_{s2}⨁{\mathrm{SS}}_{k}\;\right)\approx U^{*}=H\left(Y⨁{\mathrm{SS}}_{k}⨁T_{s3}\right)$


B) Initial Assumption

In this part, the important assumptions of proposed hash-based protocol are listed that are defined as follows:
(1)$T_{R}|\equiv {\;T}_{R}\;\overset{I_{D},I_{D_{k}},{\mathrm{SS}}_{k}}{\Leftrightarrow }{\;B}_{S}$(2)$T_{R}|\equiv {\;T}_{R}\;\overset{X,Y,Z}{\Leftrightarrow }{\;B}_{S}$(3)$R_{H}\Rightarrow U$, $R_{H}|\equiv \;\#\left(U\right)$, $R_{H}\left|\equiv {\;T}_{R}\right|\equiv {\;B}_{S}\overset{U}{↔}{\;R}_{H}$(4)$R_{H}|\equiv {\;R}_{H}\overset{B_{S}}{\Leftrightarrow }$, $B_{S}\;|\equiv {\;B}_{S}\;\overset{R_{H}}{\Leftrightarrow }{\;R}_{H}$


C) Providing Security Features

In this part, three security features are provided to validate user entities and data synchronization. The detailed user entities are as follows:
(a)$B_{S}\left|\equiv {\;R}_{H}\right|\equiv {\;B}_{S}$, $R_{H}\left|\equiv B_{S}\right|\equiv {\;R}_{H}$(b)$T_{R}\left|\equiv B_{S}\right|\equiv \left\{I_{D},{\mathrm{SS}}_{k}\right\}$, $B_{S}\left|\equiv T_{R}\right|\equiv {\;B}_{S}$(c)$T_{R}|\equiv {\;B}_{S}\overset{{\mathrm{New}}_{\mathrm{msg}},T_{\mathrm{sNew}}}{↔}T_{R}$


D) Proof of Security Process

The process of security proof explains the representation of A⊢B, where A is the premise, B is the conclusion and ⊢ is a symbol of meta-linguistic. From the BAN logic, the seeing rule and the (3) protocol explanation is applied to obtain:$$R_{H}⊲\;\left\{{\left\{\left({\mathrm{SS}}_{k}∥I_{D_{K}}\right)\;⨁R_{i}\right\}}_{R_{i}},R_{i},{\left\{T_{s}\right\}}_{X,Z}\right\}⊢B_{S}\;⊲\left\{{\left\{\left({\mathrm{SS}}_{k}∥I_{D_{K}}\right)\;⨁R_{i}\right\}}_{R_{i}}\right\}$$

From the BAN logic, the freshness rule and the (4) protocol explanation are combined to attain: $$B_{S}⊲\;\left\{{\left\{\left({\mathrm{SS}}_{k}∥I_{D_{K}}\right)\right\}}_{H}\right\},{\;B}_{S}|\equiv \left(R_{i}\right)⊢B_{S}\#\left({\mathrm{SS}}_{k}∥I_{D_{K}}\right)$$

In case of $R_{H}\left|\equiv T_{R}\right|\equiv B_{S}\overset{R_{i}}{↔}R_{H}$ i.e., from (3) protocol explanation, it can be obtained as:(1)$$R_{H}\left|\equiv T_{R}\right|\equiv R_{H}$$

Similarly, it can be expressed as:(2)$$B_{S}\left|\equiv R_{H}\right|\equiv B_{S}$$

As a result, the first proven goal 〈a〉 is achieved from the given Equations (1) and (2). From the BAN logic, the seeing rule, the (2) and the (3) initial assumptions can be obtained as:$$\begin{array}{c} T_{R}\left|∼\left\{{\left\{\left(I_{D_{K}}∥R_{i}\right)⨁R_{i}\right\}}_{X},{\left\{X⨁\left(I_{D_{K}}∥R_{i}∥R_{j}\right)\right\}}_{Y},{\left\{X⨁\left(I_{D_{K}}∥T_{s1}∥R_{i}\right)\right\}}_{Z}\right\},{\;T}_{R}\right|\\ \;\;\;\;\;\;\;\equiv B_{S}\overset{X,Y,Z}{↔}T_{R} \end{array}$$

$$⊢{\;B}_{S}⊲\left\{\left\{\left(I_{D_{K}}∥R_{i}\right)⨁R_{i}\right\},\left\{X⨁\left(I_{D_{K}}∥R_{i}∥R_{j}\right)\right\},\left\{X⨁\left(I_{D_{K}}∥T_{s1}∥R_{i}\right)\right\}\right\}$$

From the initial assumption (1), it can be obtained as:(3)$$T_{R}\left|\equiv B_{S}⟹\left\{X,Z\right\},B_{S}\right|\equiv \#\left(T_{S},I_{D_{K}}\right)\;⊢{\;T}_{R}\left|\equiv B_{S}\right|\equiv \left\{T_{S},I_{D_{K}}\right\}$$

Similarly, it can be expressed as:(4)$$B_{S}\left|\equiv T_{R}\right|\equiv B_{S}$$

As a result, the second proven goal 〈b〉 is realized from the given Equations (3) and (4). According to the rule of message-meaning and the (2) initial assumption, it can be expressed as:$$T_{R}\left|\equiv B_{S}\overset{X,Y,Z}{↔}T_{R},{\;B}_{S}⊲\left\{{\left\{{\mathrm{New}}_{\mathrm{msg}},T_{\mathrm{sNew}},R_{i}\right\}}_{X,Z}\right\}⊢{\;B}_{S}\right|\equiv {\;T}_{R}\;\sim \;\left\{{\mathrm{New}}_{\mathrm{msg}},T_{\mathrm{sNew}},R_{i}\right\}\;$$

Therefore, it can be represented as:(5)$$T_{R}\left|\equiv B_{S}\overset{X,Y,Z}{↔}T_{R},B_{S}\right|⟹\left\{{\mathrm{New}}_{\mathrm{msg}},T_{\mathrm{sNew}},R_{i}\right\}⊢T_{R}|\equiv B_{S}\overset{{\mathrm{New}}_{\mathrm{msg}},T_{\mathrm{sNew}}}{↔}{\;T}_{R}$$

Similarly, it can be expressed as:(6)$$B_{S}|\equiv T_{R}\overset{{\mathrm{New}}_{\mathrm{msg}},T_{\mathrm{sNew}}}{↔}{\;B}_{S}$$

As a result, the third proven goal 〈c〉 is realized from the given Equations (5) and (6). 

From the above Equations (1) to (6), the security features 〈a〉, 〈b〉 and 〈c〉 have claimed to be successfully verified. This ensures that the proposed hash-based protocol achieves the property of mutual authentication between the tag, the reader and the back-end server. Moreover, it ensures the security of data synchronization for the proposed hash-based protocol. 

### 5.2. Informal Analysis

The proposed novel hash-based RFID mutual authentication protocol claims that it can provide a high-secure authentication against the most of the potential attacks, namely replay, eavesdropping and man-in-the-middle; since the proposed mechanism is based on the hashing operation and secret session-key synchronization: 

*Mutual Authentication*: The proposed novel hash-based RFID mechanism offers bilateral authentication between the communication parties. The back-end server authenticates the tag by the computation of $Z^{*}=H\left(A⨁H\left(I_{D_{K}}∥R_{i}∥R_{j}^{*}∥\right)\;⨁{\;R}_{j}^{*}\;⨁{\;T}_{s2}\right)$ on the server-side which will be validated with the received response-message Z sent by the reader. Correspondingly, the back-end server validates its authentication by the computation of $U^{*}=H\;\left(Y\;⨁{\;\mathrm{SS}}_{k}\;⨁{\;T}_{s3}\right)$ on the tag-side which will be identical with the received message U sent by the reader.

*Resilient to Eavesdropping and Tracing*: The proposed novel hash-based RFID mechanism is resilient to eavesdropping and tracing. As the proposed mechanism uses the random integers $R_{i}{\;\mathrm{and}\;R}_{j}$ and user anonymity, the threat of tracing can be prevented, so none of the messages transport the tag information twice owing to the challenge—response mechanism used by the independent session variables $R_{i}{\;\mathrm{and}\;R}_{j}$. The proposed mechanism has limited hash H(.) and X-OR ⨁ operations with random integers and synchronized-secret operation, therefore any threat of eavesdropping can be avoided by concealing the tag information. Hence, the proposed mechanism can successfully pass a one-way authentication step to prevent the threats, such as eavesdropping and tracing.

*Resilient to Replay Attacks*: The proposed novel hash-based RFID mechanism can be resilient to replay attacks. As the proposed mechanism often uses the random integers $R_{i}{\;\mathrm{and}\;R}_{j}$, the authentication request is verified using $U^{*}=H\;\left(Y\;⨁{\;\mathrm{SS}}_{k}\;⨁{\;T}_{s3}\right)\approx U$ to validate whether the current timestamp $T_{s3}$ is fresh or not. After the successful validation, the confidential information will be updated within the valid time-frame. Hence, the attackers can’t deduce the information shared between the reader and the tag.

*Resilient to Man-in-the-Middle Attacks*: As the proposed novel hash-based RFID mechanism uses the hashing and X-OR operations $U=H\left(X\;⨁\;H\left(I_{D_{K}}∥R_{i}∥R_{j}\right)\;⨁{\;T}_{s2}\;⨁{\;\mathrm{SS}}_{k}\right)$ for each message transmission, the parameters ${\mathrm{SS}}_{k},{\;R}_{i}{\;\mathrm{and}\;R}_{j}$ can’t be tampered with to deduce the confidential information of the tag. Hence, the proposed mechanism can be resilient to man-in-the-middle attacks. 

*Untraceability and Tag-Anonymity*: To hide the tag information, each transaction message and update process consists of some internal parameters ${\mathrm{SS}}_{k},{\;A\;\mathrm{and}\;I}_{D_{K}}$ and also uses the random integers $R_{i}{\;\mathrm{and}\;R}_{j}$. The server can deduce the identification of the tag after the successful computation of these parameters ${\mathrm{SS}}_{k},{\;A\;\mathrm{and}\;I}_{D_{K}}$ sent by the tag. Hence, the proposed mechanism stops an attacker from tracing the information of the tag as well as the secret session key of the communication system. 

*Resilient to Desynchronization Attacks*: For the successful launch of desynchronization, an attacker must be able to deduce the secret session key using the related parameters ${\mathrm{SS}}_{k}$ and ${\mathrm{SS}}_{k+1}$. In the proposed novel hash-based RFID mechanism, an attacker can’t infer the updated value of ${\mathrm{SS}}_{k}$ as it is associated with the validation of $Z^{*}=H\left(A\;⨁\;H\left(I_{D_{K}}∥R_{i}∥R_{j}^{*}∥\right)\;⨁{\;R}_{j\;}^{*}⨁{\;T}_{s2}\right)\approx Z$.

Hence, the proposed mechanism can be resilient to de-synchronization attack. Table 3 compares the security properties of various hash-based RFID protocols in which the proposed hash-based protocol is proven to be well-secured in comparison with the other hash-based RFID protocols [12,24,25]. Besides, the proposed hash-based protocol meets all the security level of context-aware sensor management systems.

### 5.3. Performance Analysis

In this subsection, the proposed hash-based RFID authentication protocol has been cross-verified with other existing authentication protocols [12,24,25]. To find the computation cost of the authentication protocols, a microcontroller family known as MSP430 simulates the SHA-256 with a frequency of 8 MHz [59]. For SHA-256, this simulator executes the hash function $T_{\mathrm{hash}}$ with 0.65 ms. From Table 4, it is observed that the proposed hash-based protocol requires 0.44 ms to complete seven hashing functions. However, Kim et al.’s [24,25] method needs 0.45 ms and 0.61 ms and that of Hajny et al. [12] requires 0.57 ms to complete the execution. Eventually, the proposed hash-based protocol consumes less computation cost in comparison with other existing schemes [12,24,25]. 

Note that Gope et al.’s method [65] requires a computation time of 0.91 ms to execute $14{\;T}_{\mathrm{hash}}$ functions, which is much more expensive than the proposed hash-based protocol. In view of communication phase interaction, i.e., forward and backward channels, the proposed hash-based protocol finishes the authentication mechanism with less interaction in comparison with other existing protocols [12,24,25]. From Table 4, it is observed that the proposed hash-based protocol invokes three interactive flows between the tags, the reader and the back-end server, whereas Kim et al. [24,25] and Hajny et al. [12] process four and seven flows, respectively. Moreover, the results prove that the proposed hash-based protocol has less communication cost as compared to other existing protocols [12,24,25], improving the system efficiency. As a result, it is claimed that it can be preferably deployed in IoT-based multimedia systems. 

## 6. Experimental Study

A powerful network simulator known as NS-3 [66] is chosen to simulate the discrete events that construct a modern network environment to investigate the communication metrics such as packet delivery ratio, end-to-end delay and throughput rate. Moreover, this tool uses both IP and non-IP to simulate the network models such as LTE-A, LTE, WiMAX etc. Importantly, it is contained in several library tools to support the required simulation functions [67]. It uses Python and C++ to build or construct the program structure. Table 5 summarizes the important parameters of the NS3 simulator.

Ubuntu 16.04 LTS is preferred to construct the simulation program that is executed for 1800 s. The RFID devices are located in rectangular fashion, which equips 20 tags in a row. To realize the communication structure, the simulation has added eight consecutive rows i.e., 160 tags, where the communication distance is set to be 25 m and the distance between the device and the gateway is assumed to be 275 m Moreover, the simulation has five readers, which can randomly mobilize with a constant speed equal to 1 m/s. The transmission range of a reader and a tag is set to be 200 m and 70 m, respectively [68]. The network environment known as IEEE 802.11 is adopted to test the above scenario. The payload is set to be 48 bytes that randomly sends and receives the information for every 4 s. This scenario is tactfully constructed to examine the communication metrics namely packet delivery ratio, end-to-end delay and throughput rate, respectively. 

### 6.1. Packet Delivery (PR) Ratio

By definition, packet delivery ratio is the ratio between the number of packets sent and the number of packets received successfully by the RFID reader. Most importantly, it is crucial to examine the performance of communication networks.

From Figure 4, it is observed that the packet delivery ratio declines when the number of devices starts to increase. Even if the number of tags reaches 100, the proposed hash-based protocol records less transmission delay between the holding reader and the communication tags in comparison with other existing schemes [12,24,25]. In addition, the result reveals the drawbacks of congestion in low-power wireless environments, where the energy is highly needed to send the data packets. Significantly, it is recorded that the energy spent is dramatically increased when there is more distance between the devices. As a result, a significant value is recommended to calculate a threshold limit, whereby the server can abort a long-distance communication to achieve better performance and device lifetime.

### 6.2. End-To-End (E2E) Delay

End-to-end (E2E) delay calculates the average time taken to send and receive the packet between the readers and the tags. It can be defined as the transmission delay, where i is the number of message transmissions, ${\mathrm{PT}}_{i}^{r}{\;\mathrm{and}\;\mathrm{PT}}_{i}^{s}$ are the timestamp set for the successful packet sent and received i.e., i-th packet transmission:(7)$$E2E=\;\frac{\sum_{i=1}^{N}\left({\mathrm{PT}}_{i}^{r}-{\mathrm{PT}}_{i}^{s}\right)}{N}$$

From Figure 5, it is noted that there is a substantial delay when the number of devices starts to increase. Due to repetitive process, more devices are trying to transmit the data packets whereby the high network congestion and distance connectivity is recorded. However, the proposed hash-based scheme has less transmission delay i.e., ~0.174 s in comparison with other authentication schemes [12,24,25].

### 6.3. Throughput Rate (TR)

Throughput rate defines the successful transmissions between the server and the reader in ≈1 m/s. It can be generally calculated using the following equation, where $T_{W}$ is the complete execution time, $Q_{I}^{R}$ quantity of packet received at the given i-th interval and $L_{i}$ is the length of transmission i-th interval packet:(8)$$\mathrm{TR}=\frac{\sum \left(Q_{i}^{R}\times L_{i}\right)}{T_{W}}$$

From Figure 6, it is found that the proposed hash-based scheme shows better packet deliverability than other authentication schemes [12,24,25]. It is also evident that the throughput rate naturally declines when the packet deliver ratio is low. 

## 7. Conclusions

In the past, computing devices have evolved for smart networking that manage sensitive data securely to improve the functioning of the IoT and communication systems. As the device interacts with its own community or infrastructure to collect environmental data over open networks, a secure authentication protocol is highly needed to prevent the unauthorized access. Therefore, a hash-based RFID authentication has been presented as a novel approach for controlling the physical access to devices. It can be useful for the various context-ware sensor management systems. While fixing the device identification and verification, the proposed hash-based RFID mechanism validates the identifier privately using a synchronized secret session key value to strengthen the privacy-enhancement of CASMS. The proposed mechanism uses hashing operations and secret session-key synchronization between the backend server and tag to meet all the security levels of CASMS, namely mutual authentication, untraceability and tag-anonymity. Moreover, the proposed mechanism is resilient to attacks, such as desynchronization, replay and man-in-the-middle. From our informal security and performance analysis, the proposed hash-based protocol achieves better efficiencies than other RFID authentication protocols. Also, the simulation using NS3 shows that the proposed protocol achieves better communication metrics such as packet delivery ratio, end-to-end delay and throughput rate in comparison with other existing schemes [12,24,25]. In the future, an optimized user verification algorithm will be proposed to reduce the operational execution time of the authentication protocol.

## Figures and Tables

**Figure 1 sensors-19-03821-f001:**
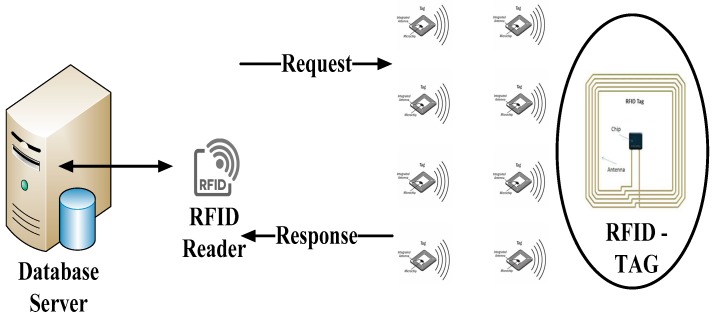
A basic system model of radio frequency identification (RFID).

**Figure 2 sensors-19-03821-f002:**
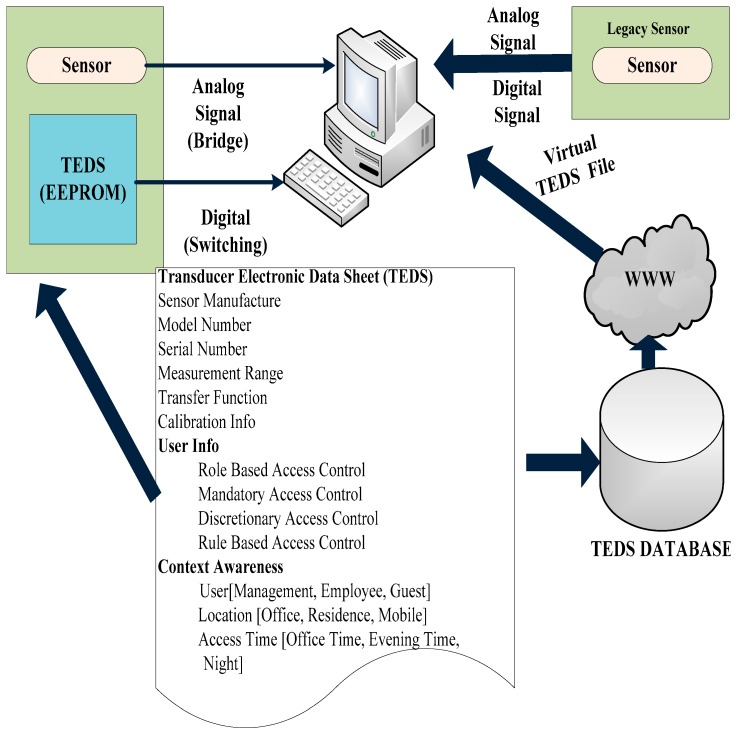
Integration of virtual TEDS systems in IEEE 1451.

**Figure 3 sensors-19-03821-f003:**
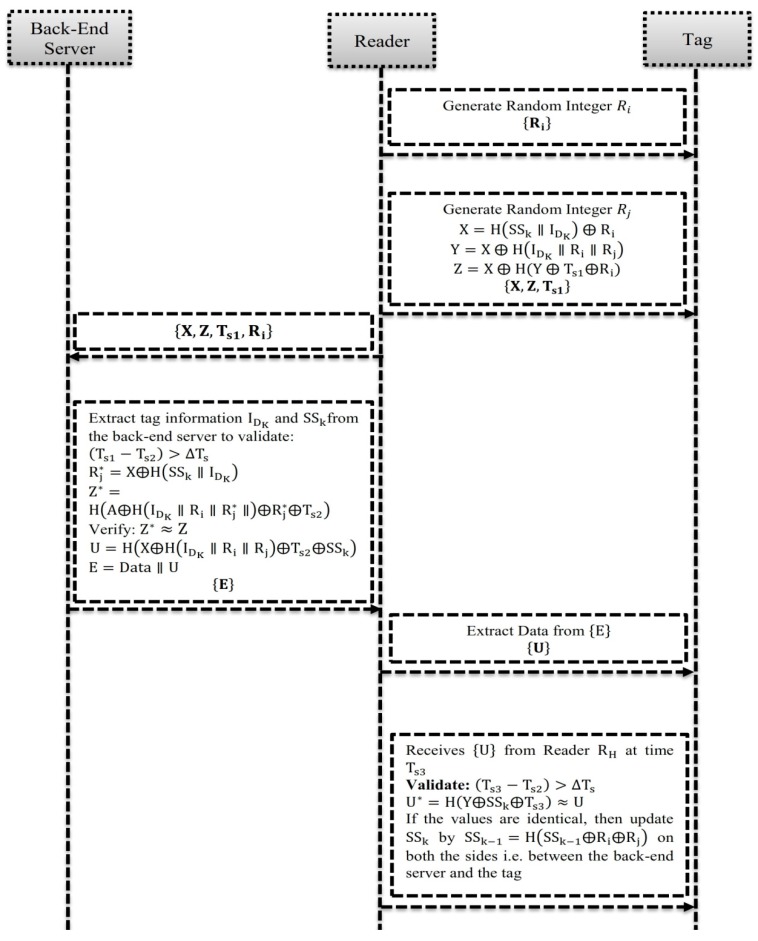
Proposed workflow of the novel hash-based RFID mechanism.

**Figure 4 sensors-19-03821-f004:**
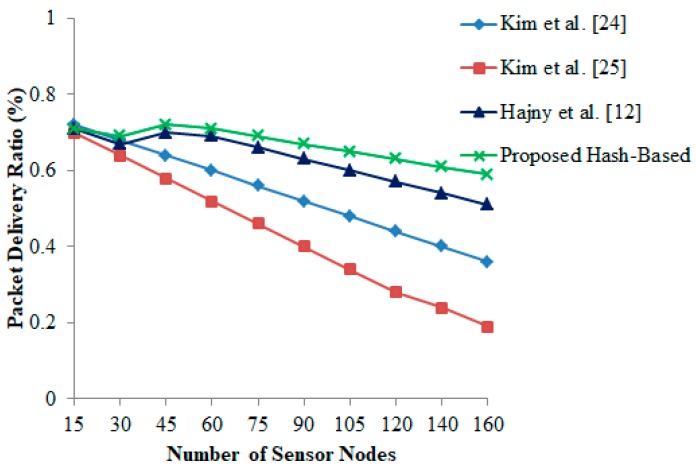
Packet delivery ratio vs. number of sensor nodes.

**Figure 5 sensors-19-03821-f005:**
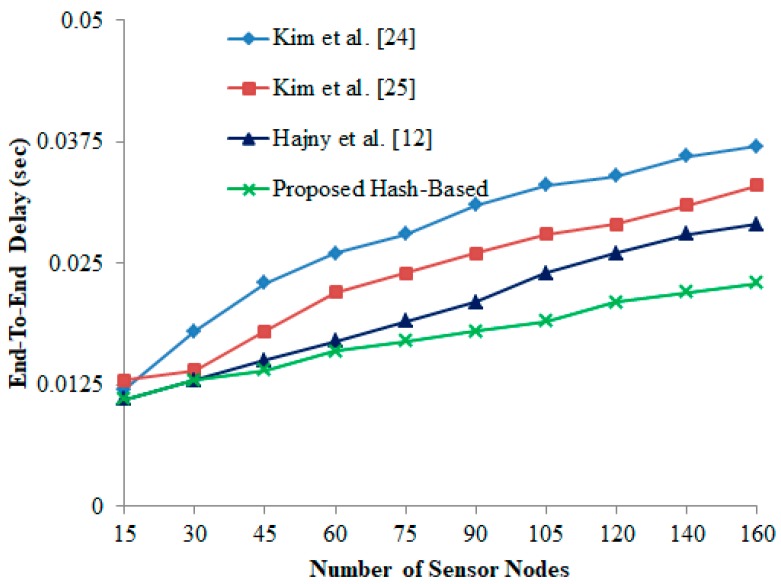
End-to-end delay vs. number of sensor nodes.

**Figure 6 sensors-19-03821-f006:**
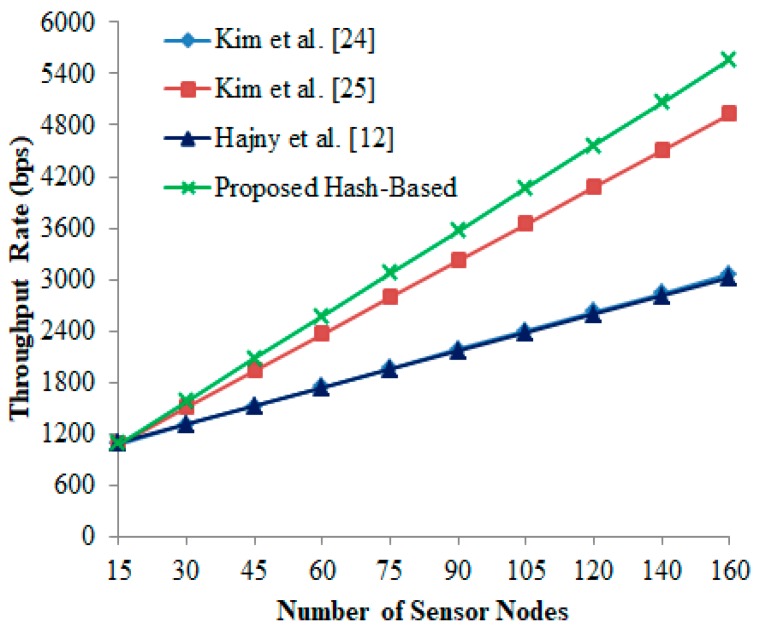
Throughput rate vs. number of sensor nodes.

**Table 1 sensors-19-03821-t001:** Challenging issues of existing RFID-based authentication protocols.

Authentication Protocol	Technique Used	Issue Addressed
Xu et al. [48]	Lightweight Authentication Using Physical Unclonable Function	Susceptible to secret disclosure and desynchronization attack
Bendavid et al. [49]	Lightweight Authentication Using Physical Unclonable Function	Perform frequent execution of setup phase to acquire a new set of pseudo-identity; whereby the back-end server experiences performance deprivation
Gope et al. [50]	Lightweight Anonymous Based Authentication Using Physical Unclonable Function	
Wang et al. [51]	Stability Guaranteed Physical Unclonable Function	
Benssalah et al. [52]	Authentication Using Elliptic Curve Signature with Message Recovery	Incur more communication cost and susceptible to untraceability

**Table 2 sensors-19-03821-t002:** Important notations used.

Notation	Description
$I_{D_{K}}$	Identity of the k-th Key
$I_{D}$	Tag identity
$R_{i}$	Random integer generated by reader
$R_{j}$	Random integer generated by tag
${\mathrm{SS}}_{k}$	Secret session-key mutually shared between back-end server and tag
${\mathrm{SS}}_{k-1}$	Secret session-key in the k-th session
H(.)	One-way hash operational function
⨁	Bitwise XOR operator
$\Delta T_{s}$	Expected transmission delay
$T_{s1}$, $T_{s2}$, $T_{s3}$	Current timestamps
∥	Concatenation operator
${\mathrm{New}}_{\mathrm{msg}}$	Message format

**Table 3 sensors-19-03821-t003:** Security properties of various hash-based RFID protocols.

Security Properties	Kim et al., 2012 [24]	Kim et al., 2013 [25]	Hajny et al. [12]	Proposed Hash-Based Protocol
Mutual Authentication	Not Support	Partial Support	Not Support	Fully Support
Resilient to Eavesdropping Attack	No	No	No	Yes
Resilient to Tracing Attack	No	No	No	Yes
Resilient to Replay Attack	No	No	No	Yes
Resilient to Man-in-the-Middle Attack	No	No	No	Yes
Resilient to De-Synchronization Attack	No	No	No	Yes
Untraceability and Tag-Anonymity	Not Provided	Not Provided	Not Provided	Provided

**Table 4 sensors-19-03821-t004:** Comparison cost of computation efficiency.

Authentication Protocol	RFID -Tag	Reader	Server	Execution Time (ms)	Communication Session	
					Forward Channel	Backward Channel
Kim et al. 2012 [24]	2 $T_{\mathrm{hash}}$	1 $T_{\mathrm{hash}}$	4 $T_{\mathrm{hash}}$	0.45	4	3
Kim et al. 2013 [25]	2 $T_{\mathrm{hash}}$	3 $T_{\mathrm{hash}}$	4 $T_{\mathrm{hash}}$	0.61	4	3
Hajny et al. [12]	2 $T_{\mathrm{hash}}$	2 $T_{\mathrm{hash}}$	4 $T_{\mathrm{hash}}$	0.57	7	4
Proposed Hash-Based Protocol	3 $T_{\mathrm{hash}}$	1 $T_{\mathrm{hash}}$	3 $T_{\mathrm{hash}}$	0.44	3	3

**Table 5 sensors-19-03821-t005:** Important parameters in NS3 Simulator.

System Parameter	Values
Operating System	Ubuntu 16.04 LTS
Simulation Time	1800 s
Area of RFID devices	$1500\times 1000{\;m}^{2}$
Availability of Readers	5 Nos.
Availability of Tags	160 Nos.
Transmission Range of a Reader	200 m
Transmission Range of a Tag	20 m
Communication Environment	IEEE 802.11
Speed of Communication device	1 m/s

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
