# Peer review of "A Hash-Based RFID Authentication Mechanism for Context-Aware Management in IoT-Based Multimedia Systems"

_sensors, 2019, doi:10.3390/s19183821_

Round 1

Reviewer 1 Report

The current paper presents an RFID authentication protocol using hash functions for IoT based systems. There is no clear novelty and significance in the paper considering numerous research studies that have already been proposed. But the main problems in such protocols where RFID are implemented is the collusion resistance of the hash functions. Especially, when the input of the hash is relatively small in the RFID protocols where commands and responses are quite small in size.

The proposed protocol does not introduces significant advances in the authentication protocols as timestamps and random numbers have been used in authentication protocols for the last 20 years.

The presentation of the paper is also poor since there is no clear related work/studies section and introductory topics are covered in the first 3 sections.

It would be better the authors to present their results in a conference rather than a journal that requires in depth security analysis with random oracles and specific attack scenarios. 

Author Response

Dear Sir/Madam,

We would like to thank the reviewers for their valuable suggestions, which were carefully read and addressed for the possible consideration. The comments and its related responses are as follows:

For Reviewer 1:

Comment 1:

The presentation of the paper is also poor since there is no clear related work/studies section and introductory topics are covered in the first 3 sections.

Response 1:

So as to provide a clear vision, we have incorporated the necessary changes in Section 1 and Section2.

You’re earnestly asked to refer Page No. 2 to Page No.5

 Comment 2:

It would be better the authors to present their results in a conference rather than a journal that requires in depth security analysis with random oracles and specific attack scenarios. 

Response 2:

So as to provide an in-depth security analysis, we have incorporated the necessary changes in Section 5 using BAN Logic.

You’re earnestly asked to refer Page No. 9 to Page No.10

Reviewer 2 Report

The structure of the article is fine, but a major revision of the language is necessary. For example:

+ Some meaningless, difficult to understand or poorly punctuated paragraphs have been detected. From the context what is meant is extracted, but the reading becomes complicated (especially in the abstract and the introduction, as they need a more general formulation). As an example, although there are many more, you can see this one:

- Lines 50-51: "... several digital meters position the large amount of RFID information..."

- Line 97: "... RFID tag physically obstructs the cryptographic operations".

- Lines 123-124: "The server such as backend database strategically operates the system processor to read..."

+ Some terms in English are not the most appropriate. Again, as an example:

- Lines 64-65: "... however it can even sense the data through the external materials"

+ There are some typos. As an example: in line 414, "optimizeds".

Paragraph 4 must be completely rewritten. It is a critical part of the article and it becomes complicated to follow the explanation. Some comments and suggestions about this are:

+ Before explaining the phases, in the paragraph beginning with line 215, the different parameters are introduced, but these are redefined in the middle of the explanation of the phases. For example, the terms SS_{k} and SS_{k-1} are defined in line 223-224 and 237-238.

+ Some notation is confusing, e.g. S_j and SS_k

+ Is it correct what is said in line 249 that {X,Z,T_{s1},R_j} is sent to the tag? Wouldn't that be the data server?

+ What is the relationship between R^*_{j} and R_{j}?

+ It is necessary to explain what each operation does and the reason for each of these. A simplified example would be helpful.

The part about results and comparison with other protocols needs some additional explanations. Specifically: 

+ Lines 345-347: Why does the proposed method have a lower computational cost than the others? 

+ The relationship of the simulation environment, explained in the paragraph beginning on line 357, with the reader-label-server structure, is not well understood. For example, who are the 5 users?

+ Some clarification is needed on what is said in lines 374-375 because the new protocol gets better results than the others.

+ For visibility reasons, it would be more appropriate to put the E2E and TR expressions outside the paragraph.

Author Response

Dear Sir/Madam,

We would like to thank the reviewers for their valuable suggestions, which were carefully read and addressed for the possible consideration. The comments and its related responses are as follows:

For Reviewer 2:

Comment 1:

- Lines 50-51: "... several digital meters position the large amount of RFID information..."

Response 1:

So as to provide better readability, we have incorporated the necessary changes in Section 1. We hope you would accept this change positively.  

‘As a result, a long-sighted RFID-tag position can easily extract a large amount of RFID information to replace traditional supply-chain management system into RFID-based authentication system.’

You’re earnestly asked to refer Page No. 2.

Comment 2:

- Line 97: "... RFID tag physically obstructs the cryptographic operations".

Response 2:

So as to provide better readability, we have incorporated the necessary changes in Section 1. We hope you would accept this change positively.  

‘Due to limited computation resource, power and storage capabilities, it is more complicated to apply the expensive cryptographic operations to a low-power RFID system. Moreover, the expensive computations have decelerated the development of RFID technologies.’

You’re earnestly asked to refer Page No. 3.

Comment 3:

- Lines 123-124: "The server such as backend database strategically operates the system processor to read..."

Response 3:

So as to provide better readability, we have incorporated the necessary changes in Section 1. We hope you would accept this change positively.  

‘The server such as backend database strategically handles the massive generated data in the views of data processing and storage to offer an influential processing capability and storage space. Furthermore, it operates the system processor to read, manage, control and store the tag data from the attached tag to the RFID reader’

You’re earnestly asked to refer Page No. 3.

Comment 4:

- Lines 64-65: "... however it can even sense the data through the external materials"

Response 4:

So as to provide better readability, we have incorporated the necessary changes in Section 1. We hope you would accept this change positively.  

‘RFID tag does not need any light source to sense the data through the external materials, which are durable, low-cost, reliable and a secure in comparison with bar-code system You’re earnestly asked to refer Page No. 2.

Comment 5:

+ There are some typos. As an example: in line 414, "optimizeds".

Response 5:

So as to provide better readability, we have incorporated the necessary changes in Section 7. We hope you would accept this change positively.  

You’re earnestly asked to refer Page No. 15.

Comment 6:

+ Before explaining the phases, in the paragraph beginning with line 215, the different parameters are introduced, but these are redefined in the middle of the explanation of the phases. For example, the terms SS_{k} and SS_{k-1} are defined in line 223-224 and 237-238.

Response 6:

So as to provide better readability, we have incorporated the necessary changes in Section 4. We hope you would accept this change positively.  

You’re earnestly asked to refer Page No.6.

Comment 7:

+ Some notation is confusing, e.g. S_j and SS_k

Response 7:

So as to provide better readability, we have incorporated the necessary changes in Section 4. We hope you would accept this change positively.  

You’re earnestly asked to refer Page No.6.

Comment 8:

+ Is it correct what is said in line 249 that {X,Z,T_{s1},R_j} is sent to the tag? Wouldn't that be the data server?

Response 8:

No, so as to make it more consistent, we have incorporated the necessary changes in Section 4. We hope you would accept this change positively.  

You’re earnestly asked to refer Page No.8.

Comment 9:

+ What is the relationship between R^*_{j} and R_{j}?

Response 9:

Random integer generated by reader

Random integer generated by tag

You’re earnestly asked to refer Page No.8.

Comment 10:

+ It is necessary to explain what each operation does and the reason for each of these. A simplified example would be helpful.

Response 10:

So as to provide better readability, we have incorporated the necessary changes in Section 4. We hope you would accept this change positively.  

Table 1 Important notations used

Notation

Description

Identity of the kth Key

Tag identity

Random integer generated by reader

Random integer generated by tag

Secret session-key mutually shared between back-end server and tag

Secret session-key in the kth session

One- way hash operational function

Bitwise  operator

Expected transmission delay

, ,

Current timestamps

Concatenation operator

Message format

You’re earnestly asked to refer Page No.6.

Comment 11:

+ Lines 345-347: Why does the proposed method have a lower computational cost than the others?

Response 11:

So as to provide better readability, we have incorporated the necessary changes in Section 4. We hope you would accept this change positively.  

‘In view of communication phase interaction i.e. forward and backward channel, the proposed hash-based protocol finishes the authentication mechanism with less interaction in comparison with other existing protocols [12,24-25]. From Table 2, it is observed that the proposed hash-based protocol invokes three interactive flows between the tags, the reader and the back-end server, whereas Kim et al. [24-25] and Hajny et al. [12] process four and seven flows respectively. Moreover, the examination results proves that the proposed hash-based protocol achieve less communication cost as compared to other existing protocols [12,24-25] to improve the system efficiency. As a result, it is claimed that it can be preferably deployed in IoT-based multimedia systems.’

You’re earnestly asked to refer Page No.6.

Comment 12:

+ The relationship of the simulation environment, explained in the paragraph beginning on line 357, with the reader-label-server structure, is not well understood. For example, who are the 5 users?

Response 12:

So as to provide better readability, we have incorporated the necessary changes in Section 6. We hope you would accept this change positively.  

Table 3 Important parameters in NS3 Simulator

System   Parameter

Values

Operating   System

Ubuntu   16.04 LTS

Simulation   Time

1800   secs

Area   of RFID devices

Availability of Readers

5 Nos.

Availability of Tags

160 Nos.

Transmission Range of a Reader

200 m

Transmission Range of a Tag

20 m

Communication Environment

IEEE 802.11

Speed   of Communication device

Ubuntu 16.04 LTS is preferred to construct the simulation program that is executed for . The RFID devices are located in rectangular fashion, which equips 20 tags in a row. To realize the communication structure, the simulation is added of eight consecutive rows i.e. 160 tags, where the communication distance is set to be  and the distance between the device and the gateway is assumed to be . Moreover, the simulation has  readers, which can randomly mobilize with the constant speed i.e. . The transmission range of a reader and a tag is set to be  and  respectively [47].

You’re earnestly asked to refer Page No.13.

Comment 13:

+ Some clarification is needed on what is said in lines 374-375 because the new protocol gets better results than the others.

Response 13:

So as to provide better readability, we have incorporated the necessary changes in Section 6. We hope you would accept this change positively.  

‘Even if the number of tags reaches to be 100, the proposed hash-based protocol records less transmission delay between the holding reader and the communication tags in comparison with other existing schemes’

You’re earnestly asked to refer Page No.13.

Comment 14:

+ For visibility reasons, it would be more appropriate to put the E2E and TR expressions outside the paragraph.

Response 13:

So as to provide better readability, we have incorporated the necessary changes in Section 6.2 and 6.3. We hope you would accept this change positively.  

You’re earnestly asked to refer Page No.14.

Round 2

Reviewer 2 Report

It seems that most of the problems indicated has been solved.

Author Response

Response To Reviewer Comments

Manuscript ID             : Sensors-555405

Type                            : Article

Authors                       : Deebak B D*, Fadi Al-Turjman , Leonardo Mostarda

Special Issue               : Internet of Multimedia Things (IoMT): Opportunities, Challenges and Solutions

We would like to thank the Guest Editor for the valuable suggestions, which were carefully read and addressed for the possible consideration again. The comments of Reviewer 1 and its related responses are as follows:

For Reviewer 1:

Comment 1:

The presentation of the paper is also poor since there is no clear related work/studies section and introductory topics are covered in the first 3 sections.

Response 1:

So as to provide a clear vision, we have incorporated the necessary changes in Section 2.3.

Table 1 Challenging Issues of Existing RFID-Based Authentication Protocols

Authentication Protocol

Technique

Used

Issue

Addressed

Xu et al. [52]

Lightweight Authentication Using Physical Unclonable Function

Susceptible to secret disclosure and desynchronization attack

Bendavid et al. [53]

Lightweight Authentication Using Physical Unclonable Function

Perform frequent execution of setup phase to acquire a new set of pseudo-identity; whereby the back-end server experiences performance deprivation  

Gope et al. [54]

Lightweight Anonymous Based Authentication Using Physical Unclonable Function

Wang et al. [55]

Stability Guaranteed Physical Unclonable Function

Benssalah et al. [56]

Authentication Using Elliptic Curve Signature with Message Recovery

Incur more communication cost and susceptible to untraceability

‘From the above discussion, the important aspects of RFID authentication protocols have been studied for IoT environments including industrial management, schedule payment, and several emergency systems. These environments use RFID-tags to achieve more computation power, speed, and physical object resistance than the traditional barcode systems. Moreover, technological advancements have addressed several security issues for RFID-based security systems. Concerning on the property of untraceability, several RFID-based authentication protocols have simply traded the privacy for the purpose of better system performance. Most of the authentication protocols merely encrypt the tag identity of RFID using cryptographic functions. In case of verification, the reader or back-end server is expected to perform an extensive operation to authorize the RFID-tag, resulting in poor system performance.

Of late, several authentication protocols have been proposed for low-cost RFID systems [44-51]. However, they are highly reported for execution cost, security weaknesses, and vulnerabilities. Chien et al. [44] presented a strong authentication and strong integrity (SASI) protocol that is based on ultra-lightweight authentication. However, their protocol is highly prone to tag tracing and desynchronization attack [45-47]. To address the critical issues of SASI, Peris-Lopez et al. [48] introduced Gossamer protocol. Unfortunately, it could not resist the desynchronization attack [49]. Of late, Fan et al. [50] presented an ultra-lightweight authentication for mobile commerce applications including the cryptographic operations such as XOR, shift and addition modulo operations. Whilst Aghili and Mala [51] proven that Fan et al. scheme cannot resist the physical and reader impersonation attack. Table 1 summarizes the challenging issues of existing RFID-based authentication protocols.

To address the above issues, this paper presents a novel hash-based RFID mechanism using synchronized secret session key value. This mechanism tactfully meets the crucial security requirements of IoT-based multimedia systems namely mutual authentication, untraceability, and resilience to desynchronization attack. Moreover, it has a built-in context aware management system to handle the storage parameters; and thus it can reduce the additional communication cost to improve the overall system performance. Importantly, it does not perform any exhaustive search to deprive the execution of back-end server.’

You’re earnestly asked to refer Page No.5

 Comment 2:

It would be better the authors to present their results in a conference rather than a journal that requires in depth security analysis with random oracles and specific attack scenarios. 

Response 2:

So as to provide an in-depth security analysis, we have incorporated the necessary changes in Section 5 using BAN Logic.

BAN Logic Analysis

BAN logic analysis is highly preferred to signify the design process that ensures the security structure of authentication protocol when it starts to build [61]. Moreover, it is a standard way to satisfy the security features of any application systems [62,63]. In general, the formal analysis model is comprised of four methods such as logical procedure, common analysis, model detection and proof of theorem. Whilst this paper chooses BAN logic to verify a belief of agreements as a logical structure [64] in order to analyze the security features of proposed hash-based protocol. The proof of verification is classified into four descriptive parts that are as follows:

Protocol Explanation

In this part, the information processes and its related transmission parameters are briefly introduced to entail the structure of the systems where  represents the RFID tag,  represents the reader and  represents the back-end server

Initial Assumption

In this part, the important assumptions of proposed hash-based protocol are listed that are defined as follows:

Providing Security Features

In this part, three security features are provided to validate user entities and data synchronization. The detailed user entities are as follows:

Proof of Security Process

The process of security proof explains the representation of , where  is the premise,  is the conclusion and  is a symbol of meta-linguistic. From the BAN logic, the seeing rule and the  protocol explanation is applied to obtain:

From the BAN logic, the freshness rule and the  protocol explanation are combined to attain:

From the above equation  to , the security features ,  and  have claimed to be successfully verified. It ensures that the proposed hash-based protocol achieves the property of mutual authentication between the tag, the reader and the back-end server. Moreover, it appeals the security of data synchronization for the proposed hash-based protocol.

You’re earnestly asked to refer Page No. 9 to Page No.11

For Reviewer 2:

Comments 1   : English language and style are fine/minor spell check required

Response 1     : As suggested, the revision draft was carefully read to correct the spell error and sentence conflict. We hope that you would satisfy with this current version.

With regards,

Deebak B D
